# Variational Assimilation of Radio Occultation Observations into Numerical Weather Prediction Models: Equations, Strategies, and Algorithms

**Michael Gorbunov** [1,2,3,*] **, Razvan Stefanescu** [1] **, Vladimir Irisov** [1] **and Dusanka Zupanski** [1]

[1] Spire Global Inc., 1825 33rd Street, Suite 100, Boulder, CO 80301, USA; razvan.stefanescu@spire.com (R.S.); vladimir.irisov@spire.com (V.I.); dusanka.zupanski@spire.com (D.Z.)

[2] A. M. Obukhov Institute of Atmospheric Physics, Russian Academy of Sciences, Pyzhevsky per. 3, Moscow 119017, Russia

[3] Hydrometeorological Research Centre of Russian Federation, B. Predtechensky per. 11-13, Moscow 123242, Russia

[*] Correspondence: gorbunov@ifaran.ru; Tel.: +7-495-951-9574

**Abstract:** We review different approaches to the variational assimilation of radio occultation (RO) observations into models of global atmospheric circulation. We derive the general equation for the bending angle that reduces to the Abel integral for a spherically layered atmosphere. We review the full 3-D observation operator for bending angles, which provides the strictest solution, but is also most computationally expensive. Commonly used is the 2-D approximation that allows treating rays as plane curve. We discuss a simple 1-D approach to the assimilation of bending angles. The observation operator based on the standard form of the Abel integral has a disadvantage, because it cannot account for waveguides. Alternative approaches use 1-D ray-tracing. The most straightforward way is to use the same framework as for the 3-D observation operator, with the refractivity field reduced to a single profile independent from the horizontal coordinates. An alternative 1-D ray-tracing approach uses the form of ray equation in a spherically layered medium that uses an invariant. The assimilation of refractivity has also 1-D and 3-D options. We derive a new simple form of the refractivity-mapping operator. We present the results of numerical tests of different 3-D and 1-D observation operators, based on Spire data.

**Keywords:** radio occultation; variational assimilation; observation operators for bending angle and refractivity

## 1. Introduction

Radio occultation observations have always been looked at as an important source of data for numerical weather prediction (NWP) [1–5]. A general approach to the use of observations in NWP is based on the variational data assimilation [6]. A way of the implementation of this approach for radio occultation (RO) observation was first discussed by Eyre [7], who applied the finite difference technique for the explicit derivation of the 3-D adjoint operator based on the ray-tracing.

Zou et al. [8] described assimilation of RO data into NWP model using the simplest observation operator: the refractivity profiles retrieved in the assumption of the spherically symmetric atmosphere were assumed to be the observables. Zou et al. [9] developed a 3-D observation operator based on the ray-tracing, and its adjoint. The bending angle was defined as the angle between the initial and final ray direction. This operator was successfully tested by Zou et al. [10] and then used by Liu et al. [11] and enhanced by Liu and Zou [12]. This approach is also referred to as 2-D approach, because it is a

good approximation to consider rays as plane curves, i.e., to neglect the transverse displacement of a ray.

Kuo et al. [13] discussed possible RO data assimilation strategies. The following candidates for observables were discussed: (1) raw phases and amplitudes in the framework of the diffraction theory; (2) raw phases in the framework of the geometrical optics (GO) under the assumption of single-ray propagation; (3) L1 and L2 bending angles; (4) ionospheric-corrected bending angle; (5) retrieved refractivity; and (6) retrieved temperature. A conclusion was made that the assimilation of bending angles, as was done by Zou et al. [10] should be preferred.

Gorbunov and Kornblueh [14] developed and implemented the strict 3-D observation operator and its adjoint. The operator is also based on ray-tracing. However, the bending angle and impact parameter are defined in terms of Doppler frequency. These are referred to as "effective" bending angle and impact parameter. The reason is that in a generic 3-D atmosphere, the ray impact parameter is not invariant for each ray, and, therefore, the impact parameters at the ray final and end points are different and cannot be derived from the observed Doppler frequency without additional assumptions. We define "effective" quantities as those that would correspond to the observed Doppler frequency, if the atmosphere were spherically symmetric. This does not involve any assumptions, because this is the definition of the observables, which is implemented in the same way in the data processing and in the observation operator and, therefore, does not result in any mismodeling. The code developed by Gorbunov and Kornblueh [14] was later modified and included into the operational data assimilation system of German Weather Service [15,16]. Although the 3-D approach is most accurate, its obvious disadvantage consists in its high computational demands.

Kursinski and Hajj [17], Healy et al. [18] followed the 1-D approach based on the retrieved refractivity. Von Engeln et al. [19] introduced a 1-D approach based on bending angles that are linked to the refractivity profile by the Abel transform. This approach was adopted by Healy and Thépaut [20], who used an approximation of the Abel integral based on piecewise exponential refractivity profile. Cucurull et al. [21–23] also used 1-D bending angle operator, but with a different numerical scheme of the evaluation of the Abel integral, based on the coordinate change eliminating the pole of the integral kernel. Burrows et al. [24] developed improvement of the interpolation scheme to reduce numerically induced biases.

Syndergaard et al. [25] introduced a refractive index mapping operator, which maps the 3-D field of model refractivity to the retrieved 1-D vertical refractivity profile. The relation between the 3-D field and 1-D profile involves the forward and inverse parts, the former representing the integration of refractivity along the rays, and the letter being the Abel inversion. Using an approximate form of ray trajectories, it was shown that it is possible to evaluate the matrix of the operator once and then use it without additional computational expenses. Sokolovskiy et al. [26,27] performed numerical simulations to test this approach with some modifications: phase observable was used instead of the retrieved refractivity. The phase observable is defined as the integral of the 1-D Abel-retrieved refractivity along straight lines. Sokolovskiy et al. [26] suggested terms "local" for 1-D operators and "non-local" for 3-D and 2-D operators. A further comparison of local and non-local operators was performed by Ma et al. [28], who confirmed that this approach allows a computationally inexpensive account of the horizontal structure using the non-local phase excess operator. Liu et al. [29] evaluated a non-local quasi-phase operator in assimilation of the CHAMP GPS refractivity using the Weather Research and Forecasting (WRF) ensemble data assimilation system. A conclusion was made that the use of the non-local RO quasi-phase operator can significantly improve the assimilation of RO data in the WRF ensemble assimilation system in the middle and lower troposphere. A further development of the mapping approach was made by Aparicio [30] who introduced the idea of using a linear or quadratic form based on the regression in order to represent the refractivity field in the vicinity of the perigee.

Healy et al. [31] investigated and compared two approaches to the 2-D bending angle observation operator. The first approach is based on the approximate form of ray and bending angle equation

represented as the Abel integral with varying impact parameter. The second approach uses the accurate ray equations in the polar coordinates. It was demonstrated that as compared to 1-D approach, 2-D observation operator has smaller modeling errors.

## 2. Basic Equations

### 2.1. Ray Equations

The electromagnetic field in a medium is described by the Maxwell equations system. Using simple approximations, it is possible to reduce the system to the following scalar (Helmholtz) equation [32]:

$$\mathcal{H}\left(\mathbf{x}, \hat{\mathbf{p}}\right) u = 0, \tag{1}$$

where $u$ is any component of the electric field, $\mathcal{H}$ is the Hamilton function, $\mathbf{x} = \left(x^i\right)$ is the vector of the Cartesian coordinates, and $\hat{\mathbf{p}} = \left(\hat{p}_i\right)$ is the covector of the momentum operators [33]:

$$\hat{p}_i = \frac{1}{\hat{i}k} \frac{\partial}{\partial x^i}, \tag{2}$$

where $\hat{i}$ stays for the imaginary unit, $k = 2\pi/\lambda$ is the wavenumber, and $\lambda$ is the wavelength. The Hamilton function has the following form:

$$\mathcal{H}\left(\mathbf{x}, \mathbf{p}\right) = \frac{1}{2}\left(\mathbf{p}^2 - n^2\left(\mathbf{x}\right)\right), \tag{3}$$

where $n\left(\mathbf{x}\right)$ is the refractivity field. The solution of the Helmholtz equation has the form of $u\left(\mathbf{x}\right) = A\left(\mathbf{x}\right)\exp\left(ik\Psi\left(\mathbf{x}\right)\right)$, where $A\left(\mathbf{x}\right)$ is the amplitude, and $\Psi\left(\mathbf{x}\right)$ is the eikonal. The lowest order asymptotic results in the eikonal equation:

$$\mathcal{H}\left(\mathbf{x}, \nabla\Psi\right) = 0, \quad n^2 = \left(\nabla\Psi\right)^2. \tag{4}$$

This a partial differential equation, whose solution can be obtained from the characteristic equations, which form the Hamilton system:

$$\dot{\mathbf{x}} = \frac{\partial\mathcal{H}}{\partial\mathbf{p}}, \quad \dot{\mathbf{p}} = -\frac{\partial\mathcal{H}}{\partial\mathbf{x}}, \quad \Psi = \mathbf{p}\dot{\mathbf{x}}, \tag{5}$$

$$\dot{\mathbf{x}} = \mathbf{p}, \quad \dot{\mathbf{p}} = n\nabla n, \quad \Psi = n^2, \tag{6}$$

where $\mathbf{p}$ is the classical momentum. Because $|\mathbf{p}| = |\nabla\Psi| = n$, as follows from (4) and (6), we arrive at the following differential relation between the parameter $t$ of system (6), the ray arc length $s$, and the eikonal:

$$dt = \frac{ds}{n}, \quad d\Psi = n\,ds. \tag{7}$$

The equation for the amplitude follows from the ray equations and the energy conservation, which, in the geometric optical approximation, requires that the energy should be transferred along rays.

Equation (6) has a form that is specific for the Cartesian coordinates. Consider an arbitrary coordinate system with the metric tensor $g_{ij}$: $ds^2 = dx^i g_{ij} dx^j$, where we, according to the rules of the tensor calculus, follow the Einstein notation implying the summation over each pair of upper and lower indexes of the same name. We define the momentum by the relation $p_i = g_{ij}\dot{x}^j$. Because form $\mathbf{p}\,d\mathbf{x}$ is invariant with respect to a coordinate change, the transform to the new coordinates $(\mathbf{p}, \mathbf{x})$ is canonical, and the canonical form of the Hamilton system also remains invariant [34], provided that the Hamilton function is defined as follows:

$$\mathcal{H}\left(\mathbf{x}, \mathbf{p}\right) = \frac{1}{2}\left(p_i g^{ij} p_j - n^2\left(\mathbf{x}\right)\right), \tag{8}$$

where $g^{ij}$ is the matrix inverse to $g_{ij}$. This results in the following form of the ray equations:

$$\dot{x}^i = \frac{\partial \mathcal{H}}{\partial p_i} = g^{ij} p_j,$$

$$\dot{p}_i = -\frac{\partial \mathcal{H}}{\partial x^i} = n\frac{\partial n}{\partial x^i} - \frac{1}{2} p_k \frac{\partial g^{kj}}{\partial x^i} p_j. \tag{9}$$

The 2-D approximation [35] allows treating rays as plane curves. Consider polar coordinates $(r, \theta)$ with the metric tensor:

$$g_{ij} = \begin{pmatrix} 1 & 0 \\ 0 & r^2 \end{pmatrix}, \quad g^{ij} = \begin{pmatrix} 1 & 0 \\ 0 & r^{-2} \end{pmatrix}. \tag{10}$$

Then we have the following equations:

$$p_\theta = r^2 \dot{\theta} = nr\frac{rd\theta}{ds} = nr \sin \psi, \tag{11}$$

$$\dot{p}_\theta = n\frac{\partial n}{\partial \theta}, \tag{12}$$

$$\dot{p}_r = \ddot{r} = n\frac{\partial n}{\partial r} + \frac{p^2}{r^3}. \tag{13}$$

where $\psi$ is the angle between ray direction $\dot{\mathbf{x}}$ and the local vertical $\mathbf{x}$. The angular component of the momentum $p_\theta$ coincides with the ray impact parameter $p$, which is invariant in a spherically layered medium. The equation for $p_\theta$ does not include $\dot{p}_r$ and can be integrated separately. Using Equation (11) and relation $ds^2 = dr^2 + r^2 d\theta^2$, we arrive at the following equation:

$$\frac{dr}{d\theta} = \pm \frac{r}{p} \sqrt{n^2 r^2 - p^2}. \tag{14}$$

Hereinafter, the upper sign relates to the ascending part of the ray trajectory from the perigee to the receiver, and the lower sign relates to descending part of the ray trajectory from the transmitter to the perigee. In the general case of a medium with horizontal gradients, this equation should be complemented with the dynamic equation for $p$. Equation (12) describes the variations of the ray impact parameter caused by horizontal gradients of refractivity [36–38].

In the perigee point, $\psi = \pi/2$ and, therefore, $p = nr$ and from Equation (13) it follows that

$$\ddot{r} = n\frac{\partial n}{\partial r} + \frac{n^2}{r} = \frac{n}{r}\frac{dx}{dr}, \tag{15}$$

where $x = nr$ is the refractive radius.

## 2.2. Bending Angle

Using the ray equations obtained above, it is straightforward to arrive at the bending angle expression. We will start with the differential equations for the bending angle:

$$d\epsilon = d\left(\psi + \theta\right), \quad \psi = \arcsin\frac{p}{nr},$$

$$d\epsilon = -d\left(\psi + \theta\right), \quad \psi = \pi - \arcsin\frac{p}{nr}, \tag{16}$$

written for the ascending and descending part of the ray trajectory, respectively. These equations can be combined as follows:

$$d\epsilon = d\arcsin\frac{p}{nr} \pm \left(\frac{dr}{d\theta}\right)^{-1} dr. \tag{17}$$

Taking into account Equation (14), we obtain the following equation:

$$
\begin{aligned}
d\epsilon &= \frac{nr}{\sqrt{n^2r^2 - p^2}}\left(\frac{dp}{nr} - \frac{p}{nr^2}dr - \frac{p}{n^2r}\left(\frac{\partial n}{\partial r}dr + \frac{\partial n}{\partial \theta}d\theta\right)\right) + \frac{pdr}{r\sqrt{n^2r^2 - p^2}} \\
&= \frac{1}{\sqrt{n^2r^2 - p^2}}\left(dp - \frac{p}{n}\left(\frac{\partial n}{\partial r}dr + \frac{\partial n}{\partial \theta}d\theta\right)\right).
\end{aligned}
\tag{18}
$$

Together with the dynamic equation for impact parameter (12), this can be transformed as follows:

$$
dp = \frac{\partial n}{\partial \theta}\frac{ds}{d\theta}d\theta = \frac{\partial n}{\partial \theta}\sqrt{\left(\frac{dr}{d\theta}\right)^2 + r^2}d\theta = \frac{\partial n}{\partial \theta}\frac{nr^2}{p}d\theta,
$$

$$
d\epsilon = \frac{1}{\sqrt{n^2r^2 - p^2}}\left(\frac{\partial n}{\partial \theta}\frac{nr^2}{p}d\theta - \frac{p}{n}\left(\frac{\partial n}{\partial r}dr + \frac{\partial n}{\partial \theta}d\theta\right)\right).
\tag{19}
$$

In addition, taking into account Equation (14) once again, we arrive at the final expression:

$$
d\epsilon = \left(-\frac{p}{n}\frac{\partial n}{\partial r} \pm \frac{\partial n}{r\,\partial \theta}\frac{\sqrt{n^2r^2 - p^2}}{n}\right)\frac{dr}{\sqrt{n^2r^2 - p^2}}.
\tag{20}
$$

At an arbitrary trajectory point, we introduce local Cartesian coordinates $(dr, rd\theta)$. The ray direction vector in this basis equals $(\cos\psi, \sin\psi)$. The expression in the brackets in Equation (20) equals the scalar product of covector $\nabla n$ and vector

$$
\left(-\frac{p}{n},\ \pm\frac{\sqrt{n^2r^2 - p^2}}{n}\right) = r\left(-\frac{p}{rn},\ \pm\sqrt{1 - \left(\frac{p}{rn}\right)^2}\right) = r\left(-\sin\psi, \cos\psi\right),
\tag{21}
$$

which is normal to the ray. In a spherically layered medium, this expression reduces to the standard formulas for the bending angle in terms of the integration over $r$ and over $x$:

$$
\epsilon\left(p\right) = -2p\int_{r_0}^{\infty}\frac{d\ln n}{dr}\frac{dr}{\sqrt{n^2r^2 - p^2}}.
\tag{22}
$$

$$
\epsilon\left(p\right) = -2p\int_{p}^{\infty}\frac{d\ln n}{dx}\frac{dx}{\sqrt{x^2 - p^2}}.
\tag{23}
$$

## 2.3. Influence of Waveguides

The dependence of the refractive radius $x\left(r\right) = n\left(r\right)r$ from the geometrical radius $r$ provides a comprehensive description of refraction in a spherically layered medium. As indicated by Equation (15), the derivative $dx/dr$ at the ray perigee point, where $\dot{r} = 0$, allows distinguishing three situations: (1) $\ddot{r} \sim dx/dr > 0$ – normal refraction (ray ascends); (2) $\ddot{r} \sim dx/dr = 0$ – critical refraction (ray slides along the spherical surface); (3) $\ddot{r} \sim dx/dr = 0$ – super-refraction (ray descends).

Because asymptotically for large height, $n\left(r\right)$ decreases exponentially, and the refractive radius becomes close to the geometrical radius, we can point out that the situation with waveguides and super-refraction corresponds to non-monotonic profiles of $x\left(r\right)$. For each ray, by virtue of Equation (11), $x \geq p$. This leads to the consideration of super-refraction illustrated in Figure 1. In some situations, to a single value of the impact parameter, multiple rays may correspond. However, only one of them can be observed from space, while the other rays are trapped by the waveguide.

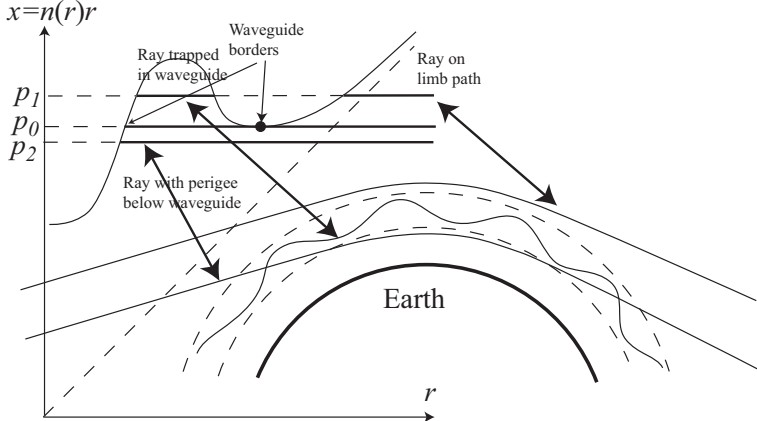

**Figure 1.** Geometry of ray propagation in the presence of a waveguide. Impact parameter $p_1$ corresponds to two rays: the first one is trapped inside the waveguide and cannot be observed from the space; the second one has a perigee above the waveguide. Impact parameter $p_2$ corresponds to a single ray with a perigee below the waveguide. Impact parameter $p_0$ corresponds to a limiting case, where there are three rays. Both the ray inside the waveguide and the ray above the waveguide asymptotically approach its upper border. The third ray slides along the upper border of the waveguide.

Note, the above derivation of the bending angle expression (22) only relied upon the fact that the ray has a single perigee point and reaches the upper border of the atmosphere. Anyway, rays with multiple perigee points must be trapped inside a waveguide and have an infinite bending angle. This expression can, therefore, be applied for the ray with impact parameter $p_2$ in Figure 1. The application of expression (23) in this situation is, however, not straightforward, because $x$ is not a unique coordinate and $n(x)$ is a multi-valued function. This integral can still be understood as a first-kind integral over a manifold, where $dx$ can change its sign. In this sense, the bending angle expression can be rewritten as follows:

$$\epsilon(p) = -2p \int_{p}^{\infty} \frac{d\ln n}{\sqrt{n^2 r^2 - p^2}}. \tag{24}$$

In solving the inverse problem, waveguide must lead to a systematic negative refractivity error [39]. To see this, consider a waveguide in the radius interval from $r_1$ to $r_2$. As follows from the above discussion, $n(r_1) r_1 = n(r_2) r_2 = x_s$. Under realistic conditions, refractivity profiles monotonically decrease with height. Assuming that $r_1 < r_2$, we see that $n(r_1) > n(r_2)$.

Consider profile $n^*(x)$, composed of profiles $n(x)$ below and above the waveguide. At the point $x_s$, the logarithm of this profile has a negative jump $\Delta \ln n = \ln n(r_2) - \ln n(r_1)$. To accurately reconstruct profile $n^*(x)$, the following bending angle profile is needed:

$$\epsilon^*(p) = \begin{cases} \epsilon(p), & p > x_s \\ -2p \int_{p}^{x_s} + \int_{x_s}^{\infty} \frac{d\ln n^*}{dx} \frac{dx}{\sqrt{x^2 - p^2}} + \frac{2p\,|\Delta \ln n|}{\sqrt{x_s^2 - p^2}}, & p < x_s \end{cases} \tag{25}$$

where the additional term accounts the delta-function in the derivative of $\ln n$ for the rays with perigee points below the waveguide. The actual observed bending angle profile for $p < x_s$ can be represented as follows:

$$\epsilon(p) = -2p \int_{p}^{x_s} + \int_{x_s}^{\infty} \frac{d\ln n^*}{dx} \frac{dx}{\sqrt{x^2 - p^2}} + 2p \int_{n_1}^{n_2} \frac{|d\ln n|}{\sqrt{x^2 - p^2}}, \tag{26}$$

where the last term relates to the piece of the profile $n$ inside the waveguide, where $x > x_s$. Therefore, we arrive at the following estimate:

$$2p \int_{n_1}^{n_2} \frac{|d \ln n|}{\sqrt{x^2 - p^2}} < \frac{2p |\Delta \ln n|}{\sqrt{x_s^2 - p^2}}. \tag{27}$$

This indicates that $\epsilon(p) < \epsilon^*(p)$ for $p < x_s$. This leads to the fact that in the presence of waveguide the Abel inversion of bending angle profiles will result in negative retrieval error of $n(r)$, while the waveguide structure cannot be retrieved.

## 3. Observation Operators

### 3.1. 3-D Assimilation of Bending Angles

#### 3.1.1. The Physical Model of RO Observations

The physical model of RO observations is based on the concept of electromagnetic waves propagating through the inhomogeneous atmosphere. However, it is not expedient to take the observed amplitudes and excess phases as observable and construct the observation operator based on the wave equation in an inhomogeneous medium. Such a formulation would be too detailed, too non-linear, and too unstable with respect to small perturbations. A much better strategy is based on the concept of the ray structure of the wave field that can be very efficiently retrieved using methods based on Fourier Integral Operators [40–46]. All these methods are closely related to each other and represent different approximations or different coordinate representations of the same solution [47]. The highest possible vertical resolution of these methods is even higher than that required by numerical weather prediction systems. Currently, the retrieval of neutral atmospheric bending angle profile is a commonly accepted procedure of the preparation of RO observation for its further assimilation. Its first step is the retrieval of bending angle profiles in the two channels, followed by the ionospheric correction [48], statistical regularization and noise reduction [49–55].

A common observation is that, due to a complicated architecture of RO experiments, any formulation of observation operator involves approximations. Given retrieved neutral bending angle, it is sufficient to formulate the forward model in terms of the relative Doppler frequency shift expressed as a function of satellite orbit data and ray geometry:

$$d = \frac{\omega_R - \omega_T}{\omega_T} = \frac{\mathbf{V}_T \cdot \mathbf{u}_T - \mathbf{V}_R \cdot \mathbf{u}_R}{c}, \tag{28}$$

where $\omega_R$ is the frequency of the received signal, and $\omega_T$ is the frequency of the transmitted signal, $\mathbf{V}_{T,R}$ are the transmitter and receiver velocities, $\mathbf{u}_{T,R}$ are unit vectors of the ray directions at the transmitter and receiver, $c$ is the light velocity in a vacuum.

The first numerical simulations of RO sounding of the Earth's atmosphere [1] involved the iterative solution of the 3-D ray boundary problem, referred to as the shooting method of locating the ray connecting two prescribed satellite positions. This led to the original formulation of the observation operator based on the excess phase $\Psi(t)$ or its normalized derivative $d(t) = (1/c)d\Psi(t)/dt$ as functions of time, together with the satellite orbit data $\mathbf{x}_{T,R}(t)$. This was thought to allow the most precise geometrical parameterization of the assimilation problem. However, the analysis of the GPS/MET RO data [56,57] indicated that multipath propagation effects play a significant role in the troposphere. This aggravates the solution of the boundary problem: (1) the model must be capable of locating multiple rays, whose number is unknown a priori; (2) there is the structural instability: at some points even small perturbations of the atmospheric state would result in the change of the number of rays; (3) the measured Doppler shift or excess phase cannot be used directly because the phase of the wave field in multipath zones is a strongly non-linear combinations of phases of the multiple

interfering rays, which need to be separated (cf. the above discussion of the observation operator based on the wave equation); (4) the measured Doppler shift and phase excess need the ionospheric correction, while the most efficient ionospheric correction is the linear combination of bending angles with the same impact parameter [48]. Finally, the solution of the multipath problem [40–46] indicated that the optimal observable to be assimilated is the bending angle as a function of the impact parameter, because the latter allows both unfolding the multipath propagation effects and reducing the ionospheric correction errors.

The bending angle is the angle between $\mathbf{u}_T$ and $\mathbf{u}_R$, and, in a general case, it cannot be determined from Equation (28), because two unit vectors have four degrees of freedom, while we have just one scalar equation. However, in the observation operator we can mimic the standard processing scheme based on the sphericity assumption [14]. In a spherically layered atmosphere, ray is a plane curve and impact parameters at the transmitter and receiver are equal to each other. Therefore, we introduce the "effective" ray directions $\tilde{\mathbf{u}}_{T,R}$ restricted to be unit vectors and satisfying Equation (28) and the above sphericity assumption:

$$\frac{\mathbf{V}_T \cdot \tilde{\mathbf{u}}_T - \mathbf{V}_R \cdot \tilde{\mathbf{u}}_R}{c} = d,$$

$$\tilde{\mathbf{u}}_T \cdot \tilde{\mathbf{u}}_T = \tilde{\mathbf{u}}_R \cdot \tilde{\mathbf{u}}_R = 1,$$

$$\mathbf{x}_R \times \tilde{\mathbf{u}}_R = \mathbf{x}_T \times \tilde{\mathbf{u}}_T, \tag{29}$$

where $\mathbf{x}_{T,R}$ are the satellite coordinates with respect to the local curvature center of the Earth's surface cross-section by the occultation plane [58,59]. This system provides six equations defining six components of vectors $\tilde{\mathbf{u}}_{T,R}$, and it is straightforward to formulate a numerical algorithm of its solution. The system solved, the effective bending angle $\epsilon$ equals the angle between $\tilde{\mathbf{u}}_T$ and $\tilde{\mathbf{u}}_R$ and the effective impact parameter $p = |\mathbf{x}_T \times \tilde{\mathbf{u}}_T| = |\mathbf{x}_R \times \tilde{\mathbf{u}}_R|$. It is important to realize that this is not an assumption or approximation. This is just the definition of the observables. Another way of expressing this is to say that the sphericity assumption errors in the observational data processing and the observation operator will mutually cancel out.

The physical model of RO observations is based on the GO ray Equation (6) complemented with the algorithm of the interpolation of gridded field of refractivity. However, this equation, by itself, allows the evaluation of a ray with the prescribed initial conditions. In a spherically layered atmosphere, this is sufficient to evaluate the impact parameter. However, in the presence of horizontal gradients, the impact parameter can only be evaluated from $\tilde{\mathbf{u}}_{T,R}$, which are function of the relative Doppler frequency shift $d$, which is a function of $\tilde{\mathbf{u}}_{T,R}$. One of the possibilities would be to solve the boundary problem, i.e., iterative solve for a ray connecting two satellites [60]. This may be conjugated with difficulties in the presence of multipath propagation, where there are multiple solutions.

We use a different approach. The observed bending angle profile $\epsilon_{\mathrm{obs}}(p)$ satisfies the following geometrical equation:

$$\theta_{RT}(t) = \epsilon_{\mathrm{obs}}(p) + \arccos\frac{p}{r_T(t)} + \arccos\frac{p}{r_R(t)}, \tag{30}$$

where $\theta_{RT}(t)$ is the angle between vectors $\mathbf{x}_T(t)$ and $\mathbf{x}_R(t)$, and $r_{T,R}(t) = |\mathbf{x}_{T,R}(t)|$. This equation can be numerically solved for the dependence of $t(p)$, which specifies the moment of time, when the ray with the impact parameter $p$ was observed. The satellite positions for this moment of time will then define the occultation plane. In this plane we locate a ray starting at the transmitter and having the impact parameter $p$. The initial approximation is the starting ray direction $\mathbf{u}_T$ satisfying the relation $p = |\mathbf{x}_T \times \mathbf{u}_T|$. In more detail we will discuss the iterative algorithm below, when discussing the adjoint version of the observation operator. Each ray is integrated to the point nearest to the receiver, because, generally speaking, the ray with effective impact parameter $p$ and starting at the transmitter

position $\mathbf{x}_T$ will, most likely, not pass through the receiver position $\mathbf{x}_R$, unless the model atmospheric state coincides with the actual atmospheric state.

### 3.1.2. Model of the 3-D Refractivity Field and Its Derivatives

To design a computational model of radio occultation experiments, a continuous model of 3-D refractivity field and its derivatives is required. In particular, the solution of the diffractive problem needs field $n(\mathbf{x})$, the numerical integration of the geometric optical ray equation needs both $n(\mathbf{x})$ and its gradient $\nabla n(\mathbf{x})$, and the linear tangent model based on the perturbation theory requires Hessian matrix $\nabla \otimes \nabla n(\mathbf{x})$. This can be done by interpolating the gridded field of the refractivity computed from the gridded fields of the specific model variables describing the atmospheric state in a specific model. For example, these variables may include temperature and humidity given at full and half sigma levels, and surface pressure, as in ECHAM models [14,61].

In a general case, there is a set of model variables $\mathbf{M}$ in the form of gridded fields of different model variables. For example, for a model with sigma levels, this vector will be represented as $\mathbf{M} = \left\{ \{P_{jk}^s\}, \{T_{ijk}\}, \{q_{ijk}\} \right\}$, where $P_{jk}^s$ is the surface pressure at the surface grid with indexes $j, k$, $T_{ijk}$ is the gridded temperature, and $q_{ijk}$ is the gridded specific humidity with index $i$ enumerating the vertical levels [61]. The surface grid may be a latitude-longitude grid $\varphi_j, \lambda_k$ or more complicated, for example an icosahedral grid $\varphi_{jk}, \lambda_{jk}$ [62]. For the sake of simplicity, we will consider latitude-longitude grids.

For any model and any grid type, it is possible to evaluate the gridded field of refractivity $n_{ijk} = n(z_{ijk}, \varphi_j, \lambda_k)$ related to geometrical grids of altitudes $z_{ijk}$, latitudes $\varphi_j$, and longitudes $\lambda_k$. This gridded field can then be interpolated to any spatial point. The interpolation procedure should be complemented with the extrapolation above the highest model grid, where we add mode altitude levels up to a height of about 120 km, and define the refractivity as a background model $n_{BG}(z, \varphi, \lambda)$ multiplied with a fitting coefficient $\alpha(\varphi, \lambda)$, chosen to minimize the difference between the model temperature and pressure and those obtained by the hydrostatic integration of the merged refractivity profile.

Because refractive index $N = n - 1$ decays with the height nearly exponentially, and for the ray integration smooth gradient of refractive index is necessary, we use the spline interpolation of $\ln N_{ijk} = \ln(n_{ijk} - 1)$ as function of $z_{ijk}$ for each vertical profile, i.e., for each fixed pair of indexes $i, j$. For given point $\mathbf{x}$ in the Cartesian coordinates, its geodetic coordinates $(z, \varphi, \lambda)$ are calculated. Then horizontal grid mesh $(\varphi_J..\varphi_{J+1}, \lambda_K..\lambda_{K+1})$ containing point $(\varphi, \lambda)$ is located. Vertically interpolated values $\ln N_{jk}(z), ln'N_{jk}(z), ln''N_{jk}(z)$ for the four pairs of indexes $j = J..J + 1$ and $k = K..K + 1$ are calculated. The linear interpolation of these values with respect to $\varphi, \lambda$-coordinates is then performed to produce $\ln N(\mathbf{x})$ and its derivatives $ln_z'N(\mathbf{x})$ and $ln_z''N(\mathbf{x})$ in the vertical direction.

Introducing the local vertical vector $\mathbf{v}(\mathbf{x}) = (\cos \varphi \cos \lambda, \cos \varphi \sin \lambda, \sin \varphi)$ in the Cartesian coordinates, we can approximately calculate the gradient and the Hessian matrix:

$$\nabla n(\mathbf{x}) = \mathbf{v}(\mathbf{x}) N(\mathbf{x}) \ln_z' N(\mathbf{x}),$$
$$\nabla \otimes \nabla n(\mathbf{x}) = \mathbf{v}(\mathbf{x}) \otimes \mathbf{v}(\mathbf{x}) N(\mathbf{x}) \left[ \ln_z'' N(\mathbf{x}) + (\ln_z' N(\mathbf{x}))^2 \right]. \tag{31}$$

The idea is to neglect the horizontal component of the gradient, which was found to be a good approximation. Taking into account the horizontal component of the gradient would require more complicated a horizontal interpolation, while the linear horizontal interpolation produces piecewise constant derivatives of $\ln N$ with respect to $\varphi, \lambda$, which are not continuous at the mesh borders.

### 3.1.3. Variations of Refractivity

The first part of the linear tangent, which is also used in the linear adjoint model, describes the variations of the refractivity due to variations of the model parameters. Since the ray trajectory

equations only include the combination $n\nabla n$, which in our model is assumed to be equal to $\mathbf{v}\langle \mathbf{v}, n\nabla n\rangle$, we only calculated the dependence of $\langle \mathbf{v}, n\nabla n\rangle$ on the model variables, i.e., derivatives:

$$\frac{\partial \langle \mathbf{v}(\mathbf{x}), n\nabla n(\mathbf{x})\rangle}{\partial \mathbf{M}} \tag{32}$$

The corresponding variations are then calculated as follows:

$$\delta \langle \mathbf{v}(\mathbf{x}), n\nabla n(\mathbf{x})\rangle = \frac{\partial \langle \mathbf{v}(\mathbf{x}), n\nabla n(\mathbf{x})\rangle}{\partial \mathbf{M}}\delta \mathbf{M} \tag{33}$$

The derivatives are evaluated by the differentiation of the interpolation scheme.

### 3.1.4. Variations of Ray Geometry

The geometric optical model is based on the numerical integration of ray trajectory Equation (6). Introducing augmented vector $\mathbf{z} = \begin{pmatrix} \mathbf{x} \\ \mathbf{u} \end{pmatrix}$ and denoting the right part of system (6) $\mathbf{F} = \begin{pmatrix} \mathbf{u} \\ n\nabla n(\mathbf{x}) \end{pmatrix}$, we rewrite the ray trajectory equation as follows:

$$\dot{\mathbf{z}} = \mathbf{F}(\mathbf{z}) \tag{34}$$

This equation is integrated numerically using finite steps. Remembering the definition of $\mathbf{F}(\mathbf{z})$, we express its operator derivatives as follows:

$$\hat{\mathbf{B}}_m^\mu \equiv \left.\frac{\partial \mathbf{F}(\mathbf{z})}{\partial \mathbf{z}}\right|_{\mathbf{z}=\mathbf{z}_{m-1}^\mu} = \begin{pmatrix} \hat{\mathbf{0}} & \hat{\mathbf{I}} \\ \nabla \otimes \nabla n^2(\mathbf{x}_{m-1}^\mu)/2 & \hat{\mathbf{0}} \end{pmatrix} \approx \begin{pmatrix} \hat{\mathbf{0}} & \hat{\mathbf{I}} \\ \nabla \otimes \nabla n(\mathbf{x}_{m-1}^\mu) & \hat{\mathbf{0}} \end{pmatrix} \tag{35}$$

where $\hat{\mathbf{0}}$ and $\hat{\mathbf{I}}$ are the zero and unit matrices of dimension $3 \times 3$, index $m$ enumerates integration steps, and index $\mu$ enumerates the substeps within each integration step, defined by the specific numerical integration scheme.

Introducing variations $\bar{\delta}\mathbf{F}_{m-1}^\mu$ of the form of the right part due to variations of the model refractivity, we can derive the following expression for variations of $\mathbf{z}_m$:

$$\delta \mathbf{z}_m = \hat{\mathbf{B}}_m \delta \mathbf{z}_{m-1} + \sum_\mu \hat{\mathbf{C}}_m^\mu \bar{\delta}\mathbf{F}_{m-1}^\mu \tag{36}$$

where matrices $\hat{\mathbf{B}}_m$ and $\hat{\mathbf{C}}_m^\mu$ are obtained by differentiating the specific numerical integration scheme and are expressed as polynomial functions of matrices $\hat{\mathbf{B}}_m^\mu$. An example of the complete evaluation of these matrices for the Runge–Kutta scheme of the fifth order can be found in [14].

Introducing notation $\alpha_{m-1}^\mu = \langle \mathbf{v}(\mathbf{x}_{m-1}^\mu), n\nabla n(\mathbf{x}_{m-1}^\mu)\rangle$ for the parameters influencing the ray geometry, and uniting parameters $\alpha_{m-1}^\mu$ into vector $\mathbf{a}_{m-1}$, we arrive at the expression for the variations of the form of the right part:

$$\bar{\delta}\mathbf{F}_{m-1}^\mu = \hat{\mathbf{A}}_m^\mu \delta \mathbf{a}_{m-1} = \hat{\mathbf{A}}_m^\mu \frac{\partial \mathbf{a}_{m-1}}{\partial \mathbf{M}}\delta \mathbf{M} \tag{37}$$

where the matrices $\hat{\mathbf{A}}_m^\mu$ cut out $\alpha_{m-1}^\mu$ from $\mathbf{a}_{m-1}$. Collecting all these transformations, we arrive at the equation for the variations of augmented vector $\mathbf{z}$:

$$\delta \mathbf{z}_m = \hat{\mathbf{B}}_m \delta \mathbf{z}_{m-1} + \sum_\mu \hat{\mathbf{C}}_m^\mu \hat{\mathbf{A}}_m^\mu \frac{\partial \mathbf{a}_{m-1}}{\partial \mathbf{M}}\delta \mathbf{M} \equiv \hat{\mathbf{B}}_m \delta \mathbf{z}_{m-1} + \hat{\mathbf{C}}_m \delta \mathbf{M} \tag{38}$$

### 3.1.5. Variations of Refraction Angle

The refraction angle and impact parameter are functions of the initial and final conditions of a ray trajectory, as defined by Equation (29):

$$
\begin{aligned}
\epsilon &= \epsilon(\mathbf{z}_0, \mathbf{z}_N) \\
p &= p(\mathbf{z}_0, \mathbf{z}_N)
\end{aligned}
\tag{39}
$$

Final condition $\mathbf{z}_N$ is a function of the initial condition and model variables:

$$
\mathbf{z}_N = \mathbf{z}_N(\mathbf{z}_0, \mathbf{M})
\tag{40}
$$

The full variations of the refraction angle and impact parameter can be written in the following form:

$$
\begin{aligned}
\delta\epsilon &= \frac{\partial\epsilon}{\partial\mathbf{z}_0}\delta\mathbf{z}_0 + \frac{\partial\epsilon}{\partial\mathbf{z}_N}\frac{\partial\mathbf{z}_N}{\partial\mathbf{z}_0}\delta\mathbf{z}_0 + \frac{\partial\epsilon}{\partial\mathbf{z}_N}\frac{\partial\mathbf{z}_N}{\partial\mathbf{M}}\delta\mathbf{M} \\
\delta p &= \frac{\partial p}{\partial\mathbf{z}_0}\delta\mathbf{z}_0 + \frac{\partial p}{\partial\mathbf{z}_N}\frac{\partial\mathbf{z}_N}{\partial\mathbf{z}_0}\delta\mathbf{z}_0 + \frac{\partial p}{\partial\mathbf{z}_N}\frac{\partial\mathbf{z}_N}{\partial\mathbf{M}}\delta\mathbf{M}
\end{aligned}
\tag{41}
$$

We need the variation of the refraction angle with a given impact parameter, so we choose the variation of the initial condition so that $\delta p$ should be equal to 0. To arrive at a completely determined system of conditions for $\delta\mathbf{z}_0$, we assume that we only vary ray direction $\mathbf{u}_0$, and that its variation is coplanar with vectors $\mathbf{x}_0$ and $\mathbf{x}_N$. Complementing this with the requirement for varied $\mathbf{u}_0$ to remain a unit vector, we can uniquely define $\delta\mathbf{z}_0$ from the following system:

$$
\begin{aligned}
&\left(\frac{\partial p}{\partial\mathbf{z}_0} + \frac{\partial p}{\partial\mathbf{z}_N}\frac{\partial\mathbf{z}_N}{\partial\mathbf{z}_0}\right)\delta\mathbf{z}_0 \equiv \frac{dp}{d\mathbf{z}_0}\delta\mathbf{z}_0 = -\frac{\partial p}{\partial\mathbf{z}_N}\frac{\partial\mathbf{z}_N}{\partial\mathbf{M}}\delta\mathbf{M} \\
&\delta\mathbf{x}_0 = 0 \\
&(\delta\mathbf{u}_0, [\mathbf{x}_0, \mathbf{x}_N]) = 0 \\
&(\mathbf{u}_0, \delta\mathbf{u}_0) = 0
\end{aligned}
\tag{42}
$$

The full derivative $\dfrac{dp}{d\mathbf{z}_0}$ is used in the iterative solution for the initial condition of ray with prescribed impact parameter $p$. We shall use the following symbolic notation for the solution of this system:

$$
\delta\mathbf{z}_0 = -\frac{d\mathbf{z}_0}{dp}\frac{\partial p}{\partial\mathbf{z}_N}\frac{\partial\mathbf{z}_N}{\partial\mathbf{M}}\delta\mathbf{M}
\tag{43}
$$

For the variation of the refraction angle, we have the following expression:

$$
\delta\epsilon = \left[\frac{\partial\epsilon}{\partial\mathbf{z}_N} - \left(\frac{\partial\epsilon}{\partial\mathbf{z}_0} + \frac{\partial\epsilon}{\partial\mathbf{z}_N}\frac{\partial\mathbf{z}_N}{\partial\mathbf{z}_0}\right)\frac{d\mathbf{z}_0}{dp}\frac{\partial p}{\partial\mathbf{z}_N}\right]\frac{\partial\mathbf{z}_N}{\partial\mathbf{a}}\delta\mathbf{a} \equiv \frac{d\epsilon}{d\mathbf{z}_N}\frac{\partial\mathbf{z}_N}{\partial\mathbf{M}}\delta\mathbf{M}
\tag{44}
$$

The symbolic full derivative $\dfrac{d\epsilon}{d\mathbf{z}_N}$ introduced here describes the sensitivity of the refraction angle with respect to the ray geometry. Equation (44) is the expression of the adjoint observation operator for the bending angle as a function of the impact parameter.

### 3.1.6. Error Covariances

The assimilation requires estimates of error covariances. The ionospheric correction algorithm combined with the statistical optimization and with the residual error estimate was developed by [51]. A simple model of covariances is described by [63]. A dynamical algorithm of the covariance estimate was introduced by [64]. Radio holographic estimates of bending angles errors were described by [65]

and further developed by [66]. More advanced analysis of uncertainty propagation through the retrieval chain and resulting covariances was performed by [53,54,67–69].

*3.2. 1-D Assimilation of Bending Angles*

3.2.1. Operators Based on the Abel Integral

The 1-D assimilation of bending angles [17–23] uses the representation of the bending angle as the Abel integral of a single profile of refractivity gradient over the refractive radius $x = nr$, as specified by Equation (23). This representation is convenient, because $\epsilon(p)$ is a linear functional of $n(x)$.

There are different numerical schemes of evaluation of the Abel integral on finite grids. Ref. [20] use a piecewise exponential representation of $N(x)$ between the nodes of grid $x_i, i = 1, ..., K$:

$$\ln n \approx n - 1 \equiv N,$$
$$k_i = \frac{\ln(N_i/N_{i+1})}{x_{i+1} - x_i},$$
$$\frac{d \ln n}{dx} \approx k_i N_i \exp(-k_i(x - x_i)), \quad x_i \leq x \leq x_{i+}, \tag{45}$$

as well as the following approximation of the integral kernel:

$$\sqrt{x^2 - p^2} \approx \sqrt{2p}\sqrt{x - p}. \tag{46}$$

This results in the following discrete approximation of the bending angle:

$$\epsilon(p) = \sqrt{2\pi p} \sum_{i=1}^{K-1} k_i N_i \exp(k_i(x_i - p)) \left[ \mathrm{erf}\left(\sqrt{k_i(x_{i+1} - p)}\right) - \mathrm{erf}\left(\sqrt{k_i(x_i - p)}\right) \right]. \tag{47}$$

Alternatively, the following coordinate change [21] is performed:

$$x = \sqrt{p^2 + s^2}, \tag{48}$$

which eliminates the pole of the integral kernel and allows the numerical integration in an equally spaced grid in $s$, using the trapezoidal rule.

The linear adjoint model uses the block for the evaluation of the derivatives of the refractivity gradient with respect to the model variables. It must be complemented with the following relationships:

$$\frac{dn}{dx} = \frac{n'_r}{n + rn'_r},$$
$$\left.\frac{\partial}{\partial \mathbf{M}} \frac{dn}{dx}\right|_x = \frac{\partial}{\partial \mathbf{M}} \frac{dn}{dx} - \frac{d^2 n}{dx^2} \frac{\partial x}{\partial \mathbf{M}},$$
$$\left.\frac{\partial n}{\partial \mathbf{M}}\right|_x = \frac{\partial n}{\partial \mathbf{M}} - \frac{dn}{dx} \frac{\partial x}{\partial \mathbf{M}}, \tag{49}$$

the two last expressions providing the derivatives for a fixed value of $x$. They are necessary, because $x = nr$ also depends on the refractivity.

These approaches based on the Abel integral over refractive radius $x$ rely upon $n(x)$ being a single-valued function. The above discussion of the waveguides in Section 2.3 indicates that for the points located below a waveguide, bending angle, although being defined, requires the integration over two branches of a multi-valued dependence $n(x)$ or the integration over the geometrical radius $r$, expressed by Equation (22). In this case, however, simple analytical formulas for the bending angle increment at each step of the grid cannot be derived due to a non-linear dependence of the sub-integral expression from $n$.

### 3.2.2. 1-D Ray-Tracing Operators

A straightforward solution for 1-D observation operator that can be applied in the presence of waveguides is the use of ray-tracing. This approach mimics the 3-D ray-tracing approach, as described in Section 3.1, with one single modification. Instead of using 3-D gridded fields of refractivity, one single 1-D vertical profile without horizontal interpolation is used. This approach will be referred to as 3D_CAR_RT, which stays for 3-D ray-tracing with constant atmospheric refractivity along horizontal. The 3-D ray-tracing taking into account the horizontal gradients of refractivity will be referred to as 3D_RT. From the viewpoint of the computational expenses, 3D_CAR_RT approach is much cheaper than 3D_RT. On the other hand, 3D_CAR_RT is not equivalent to the Abel integral, because it can simulate the realistic geometry of the Earth, including the reference ellipsoid and geoid, rather than spherical symmetry centered at the local curvature center.

Instead of using the Abel integral, it is possible to use the special form of the ray trajectory equation for spherically layered medium, from which the Abel integral follows, as shown in Section 2. We will refer to this approach as 1D_RT. The simplest equation with the order reduced by using a symmetry is Equation (14) written for a single half of the ray trajectory:

$$\frac{dr}{d\theta} = \frac{r}{p}\sqrt{n^2 r^2 - p^2}. \tag{50}$$

This equation does not contain the gradient of refractivity, which makes it numerically more stable. Due to this, 1D_RT has a smaller computational demand as compared to 3D_CAR_RT, although both these approaches integrate a differential equation depending on just one vertical profile of refractivity.

This equation, however, needs some special treatment of the perigee point: given the perigee radius $r_p$ that satisfies the equation $n(r_p) r_p = p$, Equation (50) has a constant solution $r(\theta) = r_p$. Therefore, it must be complemented with the initial condition for the second derivative. Using Equation (15) and relation $\Delta\theta = n\,\Delta t/r$ at the perigee point, we start the integration at $r(\theta = 0) = r_p$, and make the first step:

$$r(\Delta\theta) = r_p + \frac{\Delta\theta^2}{2}\frac{r}{n}\frac{dx}{dr}\bigg|_{r=r_p} = r_p + \frac{\Delta\theta^2}{2}r_p\left(1 + r_p\frac{n'}{n}\right). \tag{51}$$

After that, Equation (50) is integrated, until $r$ reaches the pre-specified upper boundary, e.g., $r_E + 120$ km, where $r_E$ is the Earth's local curvature radius. The bending angle is then evaluated as follows:

$$\epsilon = 2\theta + 2\arcsin\frac{p}{r(\theta)} - \pi, \tag{52}$$

assuming that $n(r) = 1$. The tangent linear operator follows the same guidelines as in Section 3.1, but is much simpler. First, the ray is integrated for a prescribed value of impact parameter $p$. Instead of 6 dynamic variables, there is only one. Differentiating the numerical integration scheme of Equation (50), with the initial condition:

$$\frac{\partial r_p}{\partial \mathbf{M}} = -\frac{\partial n}{\partial \mathbf{M}}\frac{r_p}{n + r_p n'_r}, \tag{53}$$

similarly to Equation (44), we write

$$\delta\epsilon = \frac{d\epsilon}{dr}\frac{\partial r(\theta)}{\partial \mathbf{M}}\delta\mathbf{M} = \frac{p}{r(\theta)\sqrt{r(\theta)^2 - p^2}}\frac{\partial r(\theta)}{\partial \mathbf{M}}\delta\mathbf{M}. \tag{54}$$

Here the dependence on the model state vector $\mathbf{M}$ only includes one vertical profile.

### 3.3. Assimilation of Refractivity

#### 3.3.1. 1-D Assimilation of Refractivity

1-D assimilation of refractivity practically does not differ from 1-D assimilation of bending angles, because refractivity profile is retrieved using the sphericity assumption and, therefore, is uniquely linked to the bending angle profile. It is only necessary to evaluate the covariances for the refractivity [70].

#### 3.3.2. Refractivity-Mapping Operators

Refractivity-mapping operators [25–30] should be looked at as the 3-D assimilation of refractivity. The basic idea of refractivity mapping is to evaluate the 1-D retrieved refractivity profile as a functional of 3-D refractivity field. Using an exact form of this functional would not bring anything new as compared to the 3-D assimilation of bending angles, in a way similar to 1-D assimilation of refractivity. The further idea is to use some approximate solutions for the ray trajectories that make this operator less computationally expensive.

The idea originates from [71], who derived the 2-D resolution kernel. The starting point of [71] was the approach used by [72] and originally developed by [73]. Equation (20) allows expressing the bending angle as an integral of the refractivity gradient multiplied with the ray normal vector over the ray. The exact evaluation of this integral requires the integration of the ray trajectory equations. However, using the straight-line approximation [25], neglecting the horizontal component of the gradient, and approximating logarithm of refractivity $\ln n$ as refractive index $n - 1 = N$ it can be written as follows:

$$\epsilon\left(p\right) = p \int_{p}^{\infty} \frac{\partial N\left(r, \pm \arccos \frac{p}{r}\right)}{\partial r} \frac{dr}{\sqrt{r^2 - p^2}}, \tag{55}$$

where $N = N\left(r, \theta\right)$ is a function of the radial coordinate $r$ and the angular coordinate $\theta$ in the occultation plane, $\theta = 0$ corresponding to the ray perigees. This expression is understood as the sum over two parts of the ray trajectory, which accounts for the missing factor of 2 in front of the integral. Hereinafter, expressions containing $\pm$ and/or $\mp$ signs are also understood in this sense. The corresponding Abel inversion has the following form:

$$\tilde{N}\left(r\right) = -\frac{1}{\pi} \int_{r}^{\infty} \frac{\epsilon\left(p\right) dp}{\sqrt{p^2 - r^2}}. \tag{56}$$

Substituting Equation (55) into the Equation (56), we arrive at the following operator composition:

$$\tilde{N}\left(r\right) = -\frac{1}{\pi} \int_{r}^{\infty} \int_{r}^{r'} \frac{\partial N\left(r', \pm \arccos \frac{p}{r'}\right)}{\partial r'} \frac{p\, dp}{\sqrt{p^2 - r^2}\sqrt{r'^2 - p^2}} dr'. \tag{57}$$

where $\tilde{N}\left(r\right)$ is the retrieved refractivity. If $N = N\left(r\right)$, we can evaluate the integral over $p$ using the following identity:

$$\int_{r}^{r'} \frac{p\, dp}{\sqrt{p^2 - r^2}\sqrt{r'^2 - p^2}} = \frac{\pi}{2}, \tag{58}$$

and arrive at the expected expression for a spherically layered medium:

$$\tilde{N}\left(r\right) = -\int_{r}^{\infty} \frac{dN\left(r'\right)}{dr'} dr' = N\left(r\right). \tag{59}$$

Changing in Equation (57) the coordinate $p$ to $\theta = \arccos (p/r')$, we can rewrite it in a simple form:

$$\tilde{N}\left(r\right) = -\frac{1}{\pi} \int\limits_{r}^{\infty} \int\limits_{0}^{\arccos \frac{r}{r'}} \frac{\partial N\left(r', \pm\theta\right)}{\partial r'} \frac{\cos\theta \, d\theta}{\sqrt{\cos^2\theta - \left(\frac{r}{r'}\right)^2}} dr'. \tag{60}$$

To make further simplifications, we represent $N\left(r', \theta\right)$ as follows:

$$N\left(r, \theta\right) = N\left(r, 0\right) + \int\limits_{0}^{\theta} \frac{\partial N\left(r, \theta'\right)}{\partial \theta'} d\theta'. \tag{61}$$

Substituting this into Equation (60) and integrating by parts (or switching the integration order), we arrive at the following expression:

$$\tilde{N}\left(r\right) = N\left(r\right) \mp \frac{1}{\pi} \int\limits_{r}^{\infty} \int\limits_{0}^{\arccos \frac{r}{r'}} \frac{\partial^2 N\left(r', \pm\theta'\right)}{\partial r' \, \partial \theta'} \int\limits_{\theta'}^{\arccos \frac{r}{r'}} \frac{\cos\theta \, d\theta}{\sqrt{\cos^2\theta - \left(\frac{r}{r'}\right)^2}} d\theta' \, dr'. \tag{62}$$

Analytically evaluating the integral over $\theta$ and changing notation from $\theta'$ to $\theta$, we finally arrive at the following expression:

$$
\begin{aligned}
\tilde{N}\left(r\right) &= N\left(r, 0\right) \mp \frac{1}{\pi} \int\limits_{r}^{\infty} \int\limits_{0}^{\arccos \frac{r}{r'}} \frac{\partial^2 N\left(r', \pm\theta\right)}{\partial r' \, \partial \theta} \arccos \frac{r' \sin\theta}{\sqrt{r'^2 - r^2}} d\theta \, dr' \\
&= N\left(r, 0\right) \mp \frac{1}{\pi} \int\limits_{0}^{\pi/2} \int\limits_{r/\cos\theta}^{\infty} \frac{\partial^2 N\left(r', \pm\theta\right)}{\partial r' \, \partial \theta} \arccos \frac{r' \sin\theta}{\sqrt{r'^2 - r^2}} dr' \, d\theta.
\end{aligned} \tag{63}
$$

This expression represents the mapping operator as the sum of the local refractivity profile and correction term depending on the horizontal gradient. This representation of resolving kernel is simpler than that derived by [71]. Although this expression was derived using the straight-line approximation, the approximation errors will only apply to non-spherical component of the refractivity field, because the spherically symmetrical component will transform identically. On the other hand, Ref [25] suggested a simple way of taking realistic ray trajectories into account. Ray trajectories can be evaluated for the first-guess refractivity field, and then new local coordinate $\theta'$ in the occultation plane can be introduced, which takes into account the ray deviation from the straight-line ray equation.

## 4. Numerical Results

In this section, we compare the performances of the three different RO operators using Spire measurements collected between 8–28 March 2018. Spire data is generated by a constellation of low Earth orbit CubeSats in contrast with the more traditional larger satellites such as METOP and COSMIC. Bias and standard deviation of relative bending angles errors are computed using 5377 Spire RO profiles. Spire data employs an equidistant vertical grid using increments of 0.2 km. Because at altitudes higher than 50 km, the profiles are seriously affected by ionospheric errors, we decided to limit the comparison up to this altitude threshold. Finally, the statistics are computed using 4 cycles per day and 6 h GFS forecasts as background.

Figures 2 and 3 show the bias and standard deviation for Spire BA data, evaluated for three different observation operators, denoted as follows: 1D_RT is 1-D ray-tracing based on (50), 3D_CAR_RT is the 3D ray-tracing with constant atmospheric refractivity along horizontal, and 3D_RT is the full 3-D ray-tracing.

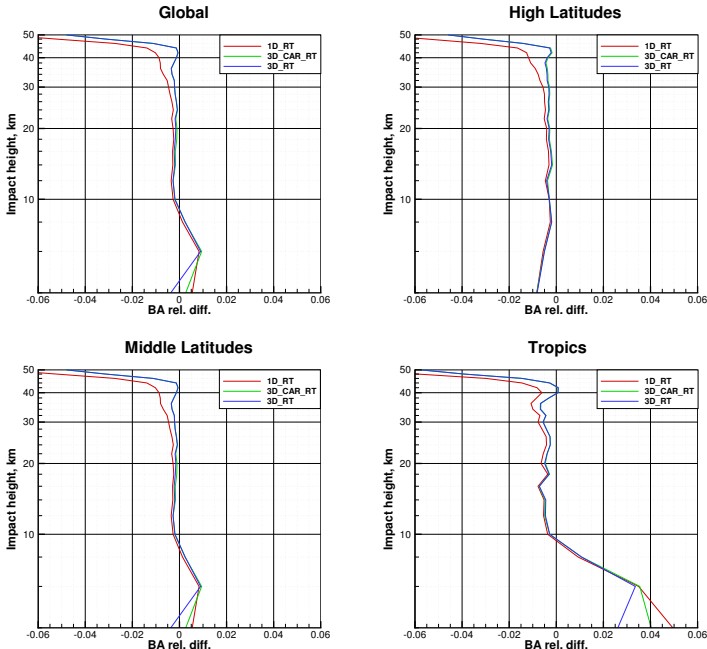

**Figure 2.** Average relative differences of Spire BA observations vs. different observation operators applied to GFS forecasts for the whole globe, high latitudes, middle latitudes, and tropics.

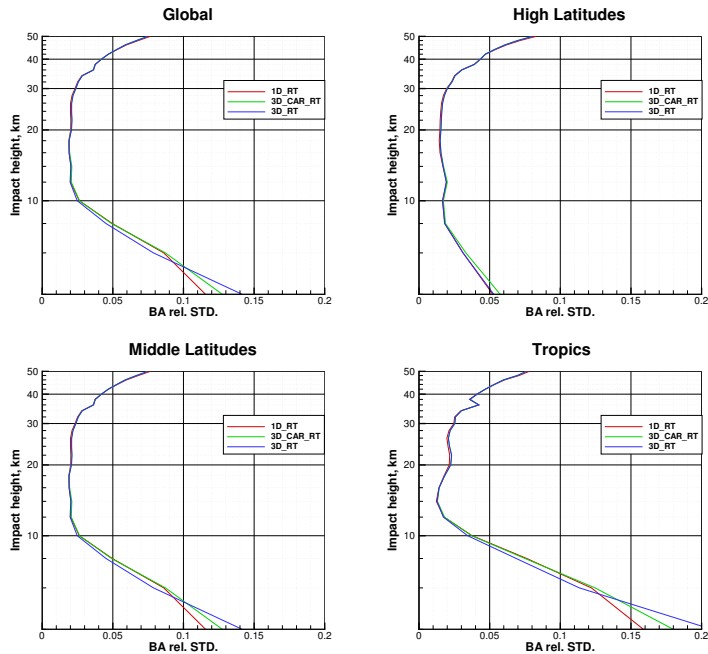

**Figure 3.** Standard deviations of relative differences of Spire BA observations vs. different observation operators applied to GFS forecasts for the whole globe, high latitudes, middle latitudes, and tropics.

## 5. Discussion

The full 3-D ray tracing observation operator, in most cases, reduces the systematic differences between GFS forecasts and RO observations in the lower troposphere, as compared to the other two simplified observation operators. The larger systematic difference for the simplest 1-D ray tracing operator at large heights come from numerical integration errors and can be reduced by using a smaller integration step. In the lower troposphere, especially in the tropics, the lowest standard deviation is

achieved for the simplest 1-D ray tracing, while for the full 3-D ray tracing it is the highest. To better understand this, a further impact study is required.

From the viewpoint of the computational expenses, 3D_CAR_RT approach is much cheaper than 3D_RT. This follows from the fact tha the highest computational expenses come from the evaluation of its derivatives with respect to the gridded fields of the model variables. Therefore, in 3D_RT, the evaluation of $\partial n/\partial \mathbf{M}$ involves much more components of the atmospheric state vector $\mathbf{M}$, as compared to 3D_CAR_RT. On the other hand, 3D_RT approach can be limited to the lower troposphere and combined with another 1-D operator above a height of 7–10 km, where the influence of the horizontal gradients upon the standard retrieval based on the Abel inversion becomes weaker.

3D_CAR_RT and 1D_RT approaches are very much equivalent in terms of computational expenses. In both cases, 1-D integration is used, and the evaluation of $\partial n/\partial \mathbf{M}$ only involve a single vertical profile of model state. Still, 3D_CAR_RT is not equivalent to the Abel integral, because it is capable of simulating the realistic geometry of the Earth, including the reference ellipsoid and geoid, rather than spherical symmetry centered at the local curvature center.

The refractivity-mapping operator (63) can be looked at as a 3-D solution that optimizes the computational costs. Although it requires the knowledge of derivatives $\partial n/\partial \mathbf{M}$ taken with respect to the full 3-D atmospheric state vector, they only need to be evaluated at a regular spatial grid, unlike the 3-D raytracing, where the derivatives are evaluated along different rays. Therefore, using an approximation of fixed occultation plane, it is possible to evaluate the necessary set of derivatives at a regular grid just once.

## 6. Conclusions

In this paper, we discussed different strategies of RO data assimilation into models of global atmospheric circulation. As was early recognized, the most convenient variable to be assimilated is the bending angle. Because the application of data processing techniques based on Fourier Integral Operators, implementing canonical transforms in the wave optics, allows the retrieval of GO bending angles, the problem can be reduced to the geometrical optics. It should also be possible to assimilate the retrieved refractivity, but it is equivalent to the assimilation of bending angles, because the refractivity is uniquely linked to the bending angle by the Abel transform. The basic equations for the bending angle can be directly derived from the wave equation, which can be solved under the GO approximation. The GO ray equations in different forms are the basis for the formulation of observation operators. When processing real RO observation, we cannot derive the bending angle and ray impact parameter, because their derivation requires an additional assumption of the spherical symmetry. Because the real atmosphere is not spherically symmetric, we introduce the concepts of the effective impact parameter and bending angle, which are defined in the way they are evaluated in the spherically symmetric case. This definition is then included into the observation operator, and, therefore, any errors due the non-sphericity of the atmosphere cancel out. We discuss different approaches to the formulation of the BA observation operator. The most efficient numerical implementation is based on the 1-D scheme, where a single atmospheric profile is used and the atmosphere is assumed to be spherically symmetric with respect to the local curvature radius of the reference ellipsoid. The observation operator can be based on the Abel integral, but in this case it is susceptible to errors due to wave guides. We introduce an alternative formulation based on a reduced-order ray equation, from which the Abel integral can be derived. Still, this form of observation operator can correctly take waveguides into account, because it does not use the assumption that the refractive radius is uniquely linked to the geometric radius. We discuss the most accurate observation operator based on the full 3-D ray-tracing equations. This operator is computationally expensive. However, it is possible to optimize the computation power demand by only using this operator in the lower troposphere, for heights below 7 km. The 3-D ray-tracing can be in a straightforward way reformulated as 1-D ray-tracing model. To this end, instead of using a 3-D interpolation scheme of the gridded refractivity field, a 1-D interpolation that only involves a single vertical profile, is employed. The form of the observation operator based on the 3-D

ray-tracing with constant atmospheric refractivity along the horizontal is more accurate than the 1-D operator based on the reduced form of ray trajectories in the spherically symmetric medium, because in the 3-D operator the realistic shape of the Earth is taken into account. We derived a simple analytic solution for the refractivity-mapping operator. This operator combines the numerical efficiency and the account of the horizontal gradients. We tested the three different variants of ray-tracing observation operators using the Spire CubeSat observations. The observation operator was evaluated for GFS forecast fields. It was shown that the full 3-D ray-tracing observation operator reduces the systematic differences between GFS forecasts and RO observations in the lower troposphere, as compared to the other two simplified observation operators. The larger systematic difference for the simplest 1-D ray-tracing operator occurs at large heights of Spire data, in the lower troposphere, especially in the tropics, the lowest standard deviation is achieved for the simplest 1-D ray-tracing, while for the full 3-D ray-tracing it is the highest.

**Author Contributions:** Formal analysis, methodology, M.G.; software, M.G. and R.S.; validation, R.S., V.I. and D.Z.

**Funding:** The research conducted by M. Gorbunov was partly funded by the Program of basic scientific research of state academies in the area of "Development of the methods of radio tomography of the atmosphere"—Development of the methodological basis of the monitoring of meteorological fields in the stratosphere and upper atmosphere, state registration No. AAAA-A18-118021290155-1, and by the Program 20 of the Russian Ministry of Education and Science.

**Acknowledgments:** The authors are grateful to Congliang Liu, Yueqiang Sun, Gottfried Kirchengast, and Jens Wickert, conferences chairs of the First International Workshop on Innovating GNSS and LEO Occultations and Reflections for Weather, Climate and Space Weather for the invitation to make a presentation at the Workshop and providing financial support.

**Conflicts of Interest:** The authors declare no conflict of interest.

## Abbreviations

The following abbreviations are used in this manuscript:

| | |
|---|---|
| RO | Radio Occultation |
| NWP | Numerical Weather Prediction |
| GO | Geometrical Optics |
| 1-D | One-dimensional |
| 2-D | Two-dimensional |
| 3-D | Three-dimensional |
| NCEP | National Centers for Environmental Prediction |
| METOP | Meteorological Operational satellite |
| COSMIC | Constellation Observing System for Meteorology, Ionosphere, and Climate |
| GFS | Global Forecast System at NCEP |

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
