# Peer review of "Variational Assimilation of Radio Occultation Observations into Numerical Weather Prediction Models: Equations, Strategies, and Algorithms"

_remotesensing, doi:10.3390/rs11242886_

Round 1

Reviewer 1 Report

The paper presents a thorough overview of fundamentals and methods developed to assimilate radio occultation observations into numerical weather prediction models. This review should be of particular interest to the readers aiming at improving and implementing effective robust and computationally inexpensive assimilation strategies of these specific data. Perhaps a small shortage of the paper is a very brief discussion of the results of sample data processing. However I believe that the paper can be accepted in the present form with some minor corrections: Line 83: “WRF” – abbreviation used without explication Equation (2) after line 95: The notation used is misleading because “i” denotes both index and imaginary unit; “k” is used without explanation. Equation (10) after line 97: The notation for diagonal matrix seems a bit informal. Taking into account that in this particular case matrices virtually reduce to scalars it could be preferable using a simpler and more compact traditional notation in brackets. Line 110: “critical refraction (ray descends)” – rather, “super-refraction” Figure 1: 1) Not quite clear what correspondence is indicated by the middle bidirectional arrow. 2) A misprint in the figure caption: “cannon” should be “can not” Line 178: A misprint. The first “xT,R” should be “xR” Line 195: “variable” – should be “variables” Equation (35) after line 226: using “n” for both refractivity and enumerator can be misleading Line 231: The note concerning matrix I (6 by 6 elements) looks irrelevant here. Figures 2 and 3: plot lines could be made more distinct. Line 309: “reducing” – rather “reduced” Line 332: “from the Abel integral follows” – an inconsistent fragment Line 340: “the3-D” – a space missing Line 349: “simples” – should be “simplest”

Author Response

See the attachement.

Reviewer 2 Report

The authors discuss variational assimilation of radio occultation observations into a NWP model using 1D and 3D options. The literature review is sufficient, and the bulk of the paper has derivations. Some of it can be placed in the appendix. The numerical results section can be elaborated to demonstrate the effectiveness of the approaches compared. My comments are as follows:

Major:

The abstract does not highlight the key results discussed in the paper. Key statistics should be presented showing the improvements. Figures 2 and 3 plots have the same x-axis labels while one shows mean and the other deviation. This should be made clearer in the plots. There are fewer details present on the numerical experiments, which should get more emphasis in the paper. 3D_CAR_RT and 1D_RT do not show much difference, although they may have different computational costs. This should be explained. The choice of Figure 3: It does not show much information. The authors should consider bias-based plots.

Author Response

See the attachement.

Round 2

Reviewer 2 Report

The authors have made the suggested changes.